# Unveiling the World of Circulating and Exosomal microRNAs in Renal Cell Carcinoma

**DOI:** 10.3390/cancers13215252

**Published:** 2021-10-20

**Authors:** José Pedro Sequeira, Vera Constâncio, João Lobo, Rui Henrique, Carmen Jerónimo

**Affiliations:** 1Cancer Biology & Epigenetics Group, IPO Porto Research Center (GEBC CI-IPOP), Portuguese Oncology Institute of Porto (IPO Porto), Porto Comprehensive Cancer Center (P.CCC), RISE@CI-IPOP (Health Research Network), Portuguese Oncology Institute of Porto (IPO Porto), R. Dr. António Bernardino de Almeid, 4200-072 Porto, Portugal; jose.leite.sequeira@ipoporto.min-saude.pt (J.P.S.); vera.salvado.constancio@ipoporto.min-saude.pt (V.C.); jpedro.lobo@ipoporto.min-saude.pt (J.L.); 2Master Programme in Oncology, School of Medicine & Biomedical Sciences, University of Porto (ICBAS-UP), Rua Jorge Viterbo Ferreira 228, 4050-513 Porto, Portugal; 3Doctoral Programme in Biomedical Sciences, School of Medicine & Biomedical Sciences, University of Porto (ICBAS-UP), Rua Jorge Viterbo Ferreira 228, 4050-513 Porto, Portugal; 4Department of Pathology, Portuguese Oncology Institute of Porto (IPOP), R. Dr. António Bernardino de Almeida, 4200-072 Porto, Portugal; 5Department of Pathology and Molecular Immunology, School of Medicine & Biomedical Sciences, University of Porto (ICBAS-UP), Rua Jorge Viterbo Ferreira 228, 4050-513 Porto, Portugal

**Keywords:** renal cell carcinoma, biomarkers, liquid biopsies, microRNA, extracellular vesicles, exosomes

## Abstract

**Simple Summary:**

Liquid biopsies have emerged as a new tool for early diagnosis. In renal cell carcinoma, this need is also evident and may represent an improvement in disease management. Hence, in this review we discuss the most updated advances in the assessment of miRNAs in liquid biopsies. Moreover, we explore the potential of circulating or exosome miRNAs in renal cell carcinoma to overcome the tissue biopsies limitations.

**Abstract:**

Renal cell carcinoma is the third most common urological cancer. Despite recent advances, late diagnosis and poor prognosis of advanced-stage disease remain a major problem, entailing the need for novel early diagnosis tools. Liquid biopsies represent a promising minimally invasive clinical tool, providing real-time feedback of tumor behavior and biological potential, addressing its clonal evolution and representing its heterogeneity. In particular, the study of circulating microRNAs and exosomal microRNAs in liquid biopsies experienced an exponential increase in recent years, considering the potential clinical utility and available technology that facilitates implementation. Herein, we provide a systematic review on the applicability of these biomarkers in the context of renal cell carcinoma. Issues such as additional benefit from extracting microRNAs transported in extracellular vesicles, use for subtyping and representation of different histological types, correlation with tumor burden, and prediction of patient outcome are also addressed. Despite the need for more conclusive research, available data indicate that exosomal microRNAs represent a robust minimally invasive biomarker for renal cell carcinoma. Thus, innovative research on microRNAs and novel detection techniques are likely to provide clinically relevant biomarkers, overcome current clinical challenges, and improve patient management.

## 1. Introduction

Reducing cancer mortality remains a main goal of the scientific community, in part materialized by the many efforts to develop effective biomarkers for early detection to decrease the proportion of cancers identified at late stages, which carry poor prognosis. Concerning renal cell carcinoma (RCC), early detection increases the likelihood of performing partial nephrectomy and, possibly, avoiding the need for adjuvant therapies, which have associated toxicities [1,2,3,4]. Moreover, in cases where such early detection is not possible, there are predictive biomarkers for response to therapy that may allow the use of second-line treatments in a timely manner. Moreover, prognostic biomarkers may provide information for selecting the best therapeutic strategy [5,6].

Overall, a cancer biomarker refers to any biological observation that can ideally replace and predict a clinically relevant outcome or an intermediate result that is more difficult to observe, and that might correspond to a protein, metabolite, RNA or DNA molecule, or an epigenetic alteration [7,8]. In addition, the pre-analytics, measurability and variability of a biomarker must be considered for its clinical application [7].

Over time, tumor tissue samples have always been the gold standard for diagnosis and prognostication. Nevertheless, this strategy faces relevant challenges. For instance, histological specimens only reflect the tumor composition at the time of sample taking. In addition, the limited quality and quantity of biomaterials derived from tissues may hamper accurate and reliable assessment of disease, and a biopsy may not provide a complete picture of the entire tumor landscape, which is particularly problematic in heterogeneous cancers such as RCC [9,10]. In addition, tissue biopsy sampling is an invasive and technically challenging tool, again particularly relevant in the kidney, considering its retroperitoneal topography [8,9]. To overcome these challenges of tissue biopsy sampling, liquid biopsies have emerged as alternative sources of clinically relevant information.

## 2. Liquid Biopsies

Unlike tissue biopsy sampling, liquid biopsies provide real-time feedback on the patient’s condition, in a minimally invasive and repeatable manner, increasing early diagnoses rate [11,12]. They often reflect tumor burden and the shifting molecular landscape of cancers, being optimal tools that favor the applicability of cellular and molecular therapies that depend on systematic and routine measurements of critical biomarkers [13].

Liquid biopsies involve the collection of body fluids, for example, blood, urine, spills or saliva using a minimally invasive method. They allow for the study of circulating tumor cells and DNA, tumor-educated platelets, extracellular vesicles (EVs) and cell-free RNA or microRNA (miRNA) (Figure 1) [8,9].

Each of them offers an immense potential, either together or as independent biomarkers for cancer, and importantly, each has its own advantages and limitations, many times related to sample type and preanalytical variables.

This emerging technique represents an ideal tool for early detection, subtyping, risk stratification and follow-up of cancer because it better represents tumor heterogeneity than tissue biopsies. In addition, they may aid in monitoring patients throughout specific targeted therapies, and pinpoint emergence of resistance that might entail the need for changing treatment schedule. Moreover, they might allow for circulating biomarkers assessment at various time points in a timely, cost-effective, specific and sensitive minimally invasive manner [8,10,14]. Thus, it is imperative to unveil and develop minimally invasive markers that may not only detect cancer, but also prognosticate and predict response to therapy [15,16].

## 3. Renal Cell Tumors

According to the World Health Organization, renal cancer (RC) was, in 2020, the 16th most incident and the 17th most deadly cancer, worldwide [17].

Among RC, RCC accounts for approximately 90% of all cases, and originates from epithelial cells of the nephron [18,19]. RCC is highly heterogeneous, as depicted by the multiple entities and molecular subtypes. Each histological entity has specific molecular backgrounds, stressing the need for subtype-specific biomarkers and, likewise, subtype-specific therapies [20].

In addition, discriminating RCC from oncocytoma (the most common benign tumor originating from the renal cortex) is a relevant clinical challenge, especially the distinction between oncocytoma and chromophobe RCC (chRCC), which may be particularly difficult in the case of the eosinophilic variant of the latter [21].

### 3.1. Diagnosis

In recent years, and despite the drop of mortality rate, RCC incidence has increased, which can be attributed mainly to incidental detection owing to the easier access to medical imaging performed for other reasons [22].

Early diagnosis of RCC is a challenge mainly because 70% of patients with localized disease remain asymptomatic or with mild symptoms, which is also related to retroperitoneal location of the kidney [3,4]. Hence, patients often develop symptoms only at later stages, and these may include acute or chronic flank pain, hypertension, anemia and cachexia [3].

RCC diagnosis is often a presumptive one, until histological confirmation. The role of physical examination is limited, although when a palpable abdominal mass, new-onset varicocele, or lower extremity edema are found, the patient should be evaluated by imaging for the presence of retroperitoneal neoplasia. Imaging may comprise computed tomography, abdominal ultrasound, magnetic resonance imaging (diffusion-weighted and perfusion-weighted imaging) and positron emission tomography [3,23]. Furthermore, a renal biopsy may be performed for diagnosis, although this technique remains underused [3]. Yet, the confirmation of malignancy may require specialized assessment of the nephrectomy specimen [24]. In this context, liquid biopsies may have the potential to become a central tool in RCC diagnosis, eventually sparing the need for nephrectomy in non-malignant conditions, although studies in this direction are still evolving [25,26,27].

### 3.2. Prognosis

The prognosis of RCC patients is highly dependent on histological subtype and TNM stage, among other factors. Some studies found that patients with advanced stage disease that undergo partial nephrectomy endure a better outcome and better post-surgical renal function than patients that undergo radical nephrectomy. Therefore, partial nephrectomy has become the first-choice therapeutic strategy, as it offers approximately the same survival time and superior renal function than radical nephrectomy [1,2].

Early diagnosis and treatment of RCC are important to increase global five-year survival, as inferred by the 90% survival rate for early-stage RCC compared to 13% of locally advanced or metastatic disease. However, patients with stage III that undergo nephrectomy have a survival rate above 70%. In addition, metastatic disease, advanced stage and invasion into the renal vein are predictors of poor prognosis in RCC [28,29]. According to a population-based study (2005–2009), 1 in 3 RCC patients was diagnosed with metastatic disease [6,30]. In addition, local recurrence or distant metastasis may be found in 20–40% of patients undergoing surgery [30,31].

In view of this, the prognosis of recurrence is variable, with the detection of early relapse being the main factor for patient prognosis [5,6]. Thus, early RCC detection, which may be provided by liquid biopsy techniques, will result in improved outcome.

## 4. MicroRNAs

MiRNAs are small non-coding RNAs, which can suppress gene expression at the translational level by directly targeting mRNA molecules. Moreover, they are involved in cell differentiation, growth, apoptosis, and proliferation [32,33]. MiRNA deregulation in cancer was first described in 2002 (chronic lymphocytic leukemia), and since then, it has been increasingly implicated in tumorigenesis [32,33,34,35].

MiRNAs have the ability to act as tumor suppressors or as oncogenic miRNAs, which are usually found down- or upregulated in cancer, respectively [34]. Moreover, miRNAs signatures seem to differ between cancer and normal tissues, as well as among cancer subtypes, thus representing a promising tumor biomarker in liquid biopsies [32,33].

Although several miRNA quantification techniques have been developed over the years, quantitative real-time polymerase chain reaction (qRT-PCR) has been the most widely used technology, especially due to the more disseminated know-how [36,37]. However, the emergence and progress of digital PCR may lead to improved miRNA analysis. With this technique, it is advocated that the steps of normalization to housekeeping miRNAs and preamplification can be obviated. In addition, it does not require triplicates and is easier to set cutoffs of positivity, compared with qRT-PCR [36,37,38,39].

Circulating miRNAs have been reported in many clinical contexts. In cancer, they have been described as biomarkers in several cancer models, including prostate and breast cancers, disclosing a diagnostic and predictive role [40,41,42].

### Circulating microRNAs in Renal Cell Carcinoma

In RCC, miRNAs have been assessed in serum, plasma and urine. A detailed description of miRNAs thus far reported for RCC diagnosis is presented in Table 1. The upregulated miR-21 and miR-106a, isolated from serum, is a potential diagnostic biomarker for ccRCC, disclosing 86.7% sensitivity and 70% specificity, with lower levels found in healthy donors and in ccRCC patients post-operatively, compared with ccRCC patients before surgery [43]. Thus, further studies are required to determine whether miR-106a levels may predict recurrence after surgery. Although the results were normalized to U6, recent studies indicate that some RNA species (i.e., U6, RNU6b, RNU48) are susceptible to degradation by serum RNAses; thus, normalization by these reference miRNAs is not reasonable [44]. Furthermore, a panel combining miR-141 and miR-1233 was reported as ccRCC diagnostic biomarker, with high sensitivity and specificity (100% and 73.3%, respectively), although no association was found between miRNAs levels and TNM stage, Furhman’s grade and SSIGN (Stage, Size, Grade and Necrosis) score [45].

MiR-210 has been reported as a diagnostic biomarker in several studies [46,47,48]. The most recent meta-analysis by Chen et al. reported 74% sensitivity and 76% specificity for this miRNA in detection of RCC [49]. Remarkably, in association with miR-378, sensitivity and specificity reached 80% and 78%, respectively [46]. When the two miRNAs levels were analyzed one week and three months after surgery, lower values were found compared to before the nephrectomy [46], which may signify that these miRNAs may be useful for clinical-decision making and evaluation of disease burden after nephrectomy. However, there was no correlation with Fuhrman’s grade, overall survival or histological RCC subtypes.

A minority of circulating miRNAs was reported in urine and plasma. Specifically, in urine, miR-15a was found to be upregulated in ccRCC patients and could detect ccRCC with a 98.1% sensitivity and 100% specificity, associating with tumor size [50]. However, no differences of miR-15a levels were found among RCC subtypes [50]. Notwithstanding the high sensitivity and specificity of urinary biomarkers, doubts about circulating miRNAs have emerged, since the aggressive urinary environment may lead to miRNAs instability and hamper detection [51].

All circulating and exosomal miRNA studies reported in this review are based on qRT-PCR. The greatest difficulty of qRT-PCR in the analysis of results is the normalization of data, as there is no consensus on which reference miRNA is most appropriate since some studies report that RNU6B/U6, 5s rRNA, RNU44/RNU48, for example, are not stable in body fluids and are hence not reliable means of normalization [44,47].

Recent studies describe that in kidney cancer studies, results’ normalization should be performed using miR-16a [47,52,53,54], however, a normalization with the above described RNAs has still being executed [43,44,45,48,50,55,56,57,58,59]. Furthermore, some authors do not normalize their results, performing absolute quantification instead. Nonetheless, no details are provided by them concerning the samples used as standards for the quantification [46,60].

Additionally, miRNAs have been reported as prognostic biomarkers (Table 2). In plasma samples, detection of lower miR-150 levels was significantly associated with both shorter overall and ccRCC-specific survival, and detection of higher miR-221 levels was also associated with poor overall survival in RCC [58,61]. Of note, however, downregulation of miR-150 may reflect an effect of blood cells, related to treatment or to impaired immune response, since this miRNA is highly expressed in mature B and T cells [61].

## 5. Extracellular Vesicles

Although circulating miRNAs seem promising as non-invasive or minimally invasive means to obtain diagnostic and/or prognostic information and to evaluate disease evolution, extracellular vesicles (EVs) have recently surfaced as an auspicious source of biomarkers for several diseases, including cancer. The vast majority of cells release EVs [23], which may resemble the alterations of tumor cells. Importantly, EVs can protect their cargo, such as miRNAs, thus representing a valuable resource for potential cancer biomarkers.

EVs have a membrane of variable size and diverse content, possibly containing lipids, peptides, enzymes, functional/structural proteins, mitochondrial DNA and a wide variety of RNAs (small RNAs, lncRNAs and messenger RNAs) capable of regulating virtually all cellular functions (Figure 2) [23,62,63]. EVs can be found in body fluids, such as plasma, serum and urine, since their bilipid membrane serves as protection against urine and blood circulation [23].

The most studied subpopulation of EVs are the exosomes. Yet, no consensus has been reached thus far concerning the definition and size of exosomes, representing the first difficulty when comparing/assessing studies related to miRNAs carried in EVs. Specifically, for the purposes of this review, we will follow the International Society of Extracellular Vesicles (ISEV) consensus, which recommends the use of the generic term “extracellular vesicles”. Nonetheless, several authors sustain that exosomes are EVs with a size between 50 and 150 nm [23]. Notwithstanding, exosome size has been a matter of debate across literature and scientific community [23,62,63], since the classification may lead to misinterpretation, because the size of microvesicles and apoptotic bodies lies between 100 nm–1 µm and 50 nm–5 µm, respectively [64].

EVs can be isolated by two major isolation techniques: density-based and size-based. Isolation by density can be performed by differential ultracentrifugation or density gradient ultracentrifugation [65]. Ultrafiltration, size exclusion chromatography, polymer precipitation and microfluidic based-strategies are techniques of EVs isolation by size [65].

The potential of EVs as RCC biomarkers, especially EV-containing miRNAs, has been addressed in several publications (summarized in Table 3) [66,67,68,69,70,71]. Isolated from serum EVs, miR-210 and miR-1233 have been shown as ccRCC diagnostic biomarkers with 70% and 81% sensitivity, and 62.2% and 76.0% specificity, respectively, and notably decreasing after nephrectomy [71]. However, no differences were found among TNM stages, raising the question as to whether they might reflect tumor burden. Wang and colleagues reported that miR-210 identified RCC with 82.5% sensitivity and 80.0% specificity. In this case, higher miR-210 levels were found in more advanced stages and higher Fuhrman grades, with no associations found with gender or age [53].

Regarding urine, different combinations of miRNAs, comprising miR-126-3p + miR-449a, miR-126-3p + miR-34b-5p, miR-126-3p + miR-486-5p, miR-25-3p + miR-34b-5p, miR-21-5p + miR-34b-5p and miR-150-5p + miR-126-3p, have been reported as diagnostic biomarkers for ccRCC, with 60.6%, 67.3%, 52.9%, 73.1%, 74% and 61.5% sensitivity, respectively, as well as 100%, 82.8%, 95.8%, 79.3%, 72.4% and 82.8% specificity, respectively [66]. The putative targets of these miRNAs were implicated in cell cycle regulation, tumorigenesis and angiogenesis [66]. Furthermore, downregulated miR-30c-5p has been reported as a potential diagnostic biomarker for early-stage ccRCC, with 68.57% sensitivity and 100% specificity [70].

Additionally, mRNA, proteins, lipids and lncRNA were also assessed in EVs released by RCC. Lower levels of mRNAs GSTA1, CEBPA and PCBD1 were found in EVs of ccRCC patients compared to healthy donors [67]. Expression of these biomarkers was higher in papillary and chromophobe RCC than in clear cell RCC [67]. Moreover, a panel composed by proteins CD10, MMP9, EMMPRIN, CAIX, DPEP1, DKK4, Syntenin 1 and AQP1 was shown to significantly differ between RCC and healthy subjects [69].

When EVs are isolated from plasma, a decrease of miR-26a-1-3p, miR-let-7-I, miRNA-615-3p was found, disclosing a significant association with highly aggressive metastatic disease in clear cell RCC [72] (Table 4). miR-let-7i-5p is a tumor suppressor in RCC cell lines, downregulating C-myc and its target genes. Dysregulation of this miRNA leads to 5-flouro-uracil resistance of RCC cells [72].

Additionally, lncARSR might represent a predictive biomarker and potential alternative target against sunitinib resistance, since this long non-coding RNA can be secreted by resistant cells, making sensitive cells resistant and fostering drug resistance. Inhibition of lncARSR in both orthotopic xenografts and PDX models suggests that this strategy may be used for overcoming sunitinib-resistant RCC [73] (Table 4).

Finally, the first association between lipid composition of urinary exosomes and RCC was first described by Del Boccio et al., disclosing a panel of 22 lipids which may allow for accurate diagnosis of clear cell RCC [68].

## 6. Circulating miRNAs versus Exosomal miRNAs

In recent years, circulating miRNAs and exosomal miRNAs (exomiRNAs) in liquid biopsies have been intensively studied. However, only few studies aimed to compare these two sources of miRNAs. Most authors argue that exosomal miRNAs represent a better way to analyze miRNAs since these seem to have more quantity and better quality and stability than circulating miRNAs [75]. Moreover, for urinary miRNAs, significant differences between circulating miRNAs and exomiRNAs have been reported [76,77].

Hence, the method that reaches the best accuracy and maximizes the detection of biomarkers should be established. For instance, Tian et al. explored the differences between the two sources of miRNAs obtained from plasma samples. In healthy donors, no differences between circulating miRNAs and exomiRNAs were apparent. However, in lung cancer patients, miRNAs (miR-181b-5p and miR-21-5p) were more enriched in exosomes than free in circulation [76].

Comparative studies have been performed in several cancer models. In lung adenocarcinoma, from a panel of six plasma miRNAs, only two were found upregulated in plasma exosomes [78]. In gastric cancer, miR-132-3p and miR-185-5p disclosed normal expression in serum exosomes, although these miRNAs were found upregulated in serum [79]. Notwithstanding the lack of comparative studies in RCC, considering the data collected and analyzed for this review, it seems that exomiRNAs disclose higher sensitivity and specificity than circulating miRNAs [44,45,48,49,53,71]. miR-1233 depicted an increase in sensitivity and specificity when detected in exosomes (of 4.4% and 50.5%, respectively [44,71]). In addition, for miR-210, Wang and colleagues found that the increase in sensitivity and specificity in exosomes was of 18.2% and 12.5%, respectively [53]. However, validation studies providing direct comparisons are required to draw more definitive conclusions [44,45,48,49,53,71]. Depending on the specific miRNA, its expression in circulation or exosomes is variable, and it may be better detected in one source or the other [80]. We thus recommend that the choice of method for studying miRNAs should be dependent on the specific miRNA and its biological context.

## 7. Conclusions

Overall, the reviewed supports the importance that EVs and miRNAs have as promising biomarkers for RCC, using liquid biopsies. This minimally invasive technique is likely to overcome the limitations of tissue biopsies and provide a more accurate and timely picture of the evolution of RCC. Nonetheless, future studies on EVs and miRNAs should focus more directly on clinical application, exploring the development of a more cost-effective and accurate tool for diagnosis and prognosis of RCC.

## Figures and Tables

**Figure 1 cancers-13-05252-f001:**
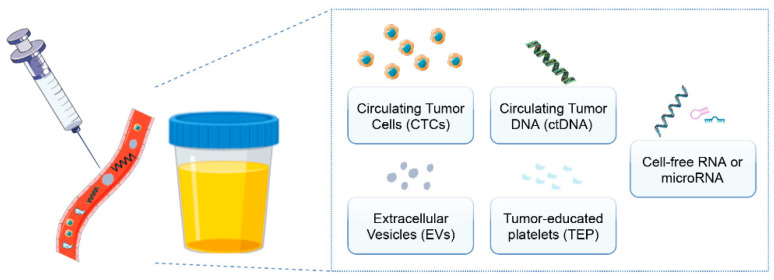
Clinical potential of liquid biopsies: what can we analyze using this promising non-invasive technique? Circulating tumor cells/DNA, extracellular vesicles, tumor-educated platelets and cell-free RNA or miRNA can be detected in the context of a liquid biopsy.

**Figure 2 cancers-13-05252-f002:**
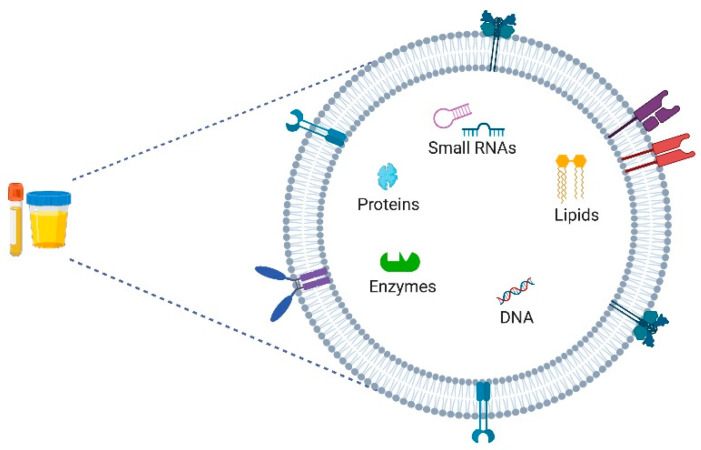
Content and cargo of extracellular vesicles: small RNAs, lipids, proteins, enzymes and DNA. Created with BioRender.com.

**Table 1 cancers-13-05252-t001:** Promising diagnostic circulating miRNA in renal cell carcinoma (RCC).

Biomarker	Clinical Application	Source	Number of Cases/Controls	Quantification Technique	Normalizer	Sensitivity %	Specificity %	AUC ^a^	REF ^b^
miR-21 ↑	Diagnostic of ccRCC	Serum	30 ccRCC ^c^ patients/30 cancer-free blood donor volunteers	qRT-PCR	U6	77.3	96.4	0.865	[43]
miR-106a ↑						86.7	70.0	0.819	
miR-34a ↓	Diagnostic of ccRCC	Serum	30 ccRCC patients (without metastatic disease)/15 non-renal benign diseases patients	qRT-PCR	Cel-miR-39	80.76	75.0	0.920	[45]
miR-141 ↓						93.33	80.0	0.780	
miR-1233 ↑						73.33	100.0	0.970	
miR-141 + miR-1233						100.0	73.3	n.a. ^d^	
miR-210 ↑	Diagnostic of ccRCC	Serum	68 ccRCC patients before surgery/42 healthy controls	qRT-PCR	5s rRNA	81.0	79.4	0.874	[48]
miR-210 ↑	Diagnostic of ccRCC	Serum	34 ccRCC patients/23 healthy controls	qRT-PCR	miR-16-5p	65.0	83.0	0.770	[47]
miR-210 ↑ + miR-378 ↑	Diagnostic of RCC ^e^	Serum	195 RCC patients (157 ccRCC, 26 pRCC ^f^, 12 chRCC ^g^)/100 healthy controls	qRT-PCR	Data not normalized	80.0	78.0	0.850	[46]
miR-378 ↑ + miR-451 ↓	Diagnostic of RCC	Serum	Screening Phase: 15 ccRCC patients/12 matched healthy controls Validation Phase: 90 RCC patients (73 ccRCC, 8 pRCC, 9 chRCC)/35 matched healthy controls	qRT-PCR	miR-16-5p	81.0	83.0	0.860	[52]
miR-1233-3p ↑	Diagnostic of RCC (No differences between RCC patients and Benign renal masses)	Serum	84 RCC patients (69 ccRCC, 10 pRCC, 3 chRCC, 2 sRCC ^h^)/93 healthy controls/13 benign renal masses	qRT-PCR	Cel-miR-39	77.4	37.6	0.588	[44]
miR-193a-3p ↑ + miR-362 ↑ + miR-572 ↑ + miR-28-5p ↓ + miR-378 ↓	Diagnostic of early-stage ccRCC	Serum	107 ccRCC patients/107 healthy controls	qRT-PCR	Let-7d/g/i	80.0	71.0	0.807	[56]
miR-210 ↑	Diagnostic of ccRCC	Serum	45 ccRCC patients/30 healthy controls	qRT-PCR	miR-16-5p	67.5	70.0	0.789	[53]
let-7a-5p ↑	Diagnostic of ccRCC	Urine	69 non-metastatic ccRCC patients/36 healthy controls	qRT-PCR	Data not normalized	71.0	81.0	0.831	[60]
miR-15a ↑	Diagnostic of RCC	Urine	67 renal tumor patients (22 ccRCC, 16 pRCC, 14 chRCC, 8 oncocytoma, 2 papillary adenoma, 5 angiomyolipoma)/15 healthy controls without kidney pathology	qRT-PCR	U6	98.1	100.0	0.955	[50]
miR-15a ↑	Diagnostic of RCC	Urine	7 ccRCC/5 chRCC/6 pRCC ^g^/5 Oncocytoma/5 Healthy Controls	qRT-PCR	5s rRNA	n.a.	n.a.	n.a.	[55]
miR-508-3p ↓	Diagnostic of RCC	Plasma	10 RCC patients/10 Healthy Controls	qRT-PCR	U6	n.a.	n.a.	n.a.	[57]

^a^ AUC—area under the curve; ^b^ REF—reference; ^c^ ccRCC—clear cell renal cell carcinoma; ^d^ n.a.—not applicable; ^e^ RCC –renal cell carcinoma; ^f^ pRCC—papilar renal cell carcinoma; ^g^ chRCC—chromophobe renal cell carcinoma; ^h^ sRCC—sarcomatoid renal cell carcinoma not otherwise specified. ↑ means upregulated and ↓ means downregulated.

**Table 2 cancers-13-05252-t002:** Promising predictive and prognostic circulating miRNA in renal cell carcinoma (RCC).

Biomarker	Clinical Application	Source	Number of Cases/Controls	Quantification Technique	Normalizer	Consequences	REF ^a^
miR-183 (high levels)	Predictive	Serum	82 RCC ^b^ patients/19 healthy controls	qRT-PCR	U6	↑ resistance to NK ^c^ cells cytotoxicity	[59]
miR-378 (high levels)	Prognostic	Serum	195 RCC patients (157 ccRCC ^d^, 26 pRCC ^e^, 12 chRCC ^f^)/100 healthy controls	qRT-PCR	Data not normalized	↓ DFS ^g^	[46]
miR-122-5p (high levels)	Prognostic	Serum	68 ccRCC/47 BRT ^h^/28 healthy controls	qRT-PCR	miR-16-5p + miR-191-5p + miR-320a	↓ OS ^i^, CSS ^j^, PFS ^k^	[54]
miR-206 (high levels)							
miR-150-5p (low levels)	Prognostic	Plasma	94 ccRCC patients/100 healthy controls	qRT-PCR	Quantile	↑ OS, DSS ^l^	[61]
miR-221-5p (high levels)	Prognostic	Plasma	43 RCC patients/34 healthy controls	qRT-PCR	RNU44	↓ OS	[58]

^a^ REF—reference; ^b^ RCC—renal cell carcinoma; ^c^ NK—natural killer; ^d^ ccRCC—clear cell renal cell carcinoma; ^e^ pRCC—papilar renal cell carcinoma; ^f^ chRCC—chromophobe renal cell carcinoma; ^g^ DFS—disease-free survival; ^h^ BRT—benign renal tumors; ^i^ OS—overall survival; ^j^ CSS—cancer-specific survival; ^k^ PFS—progression-free survival; ^l^ DSS—disease-specific survival. ↑ means longer survival and ↓ means shorter survival.

**Table 3 cancers-13-05252-t003:** Promising diagnostic exosomal miRNA in Renal Cell Carcinoma (RCC).

Biomarker	Clinical Application	Source	Number of Cases/Controls	Sensitivity %	Specificity %	AUC ^a^	Isolation	Characterization	Quantification Technique	Normalizer	REF ^b^
miR-210 ↑	Diagnostic of ccRCC ^c^	Serum	82 ccRCC patients/80 healthy controls	70.0	62.2	n.a. ^d^	Total exosome isolation reagent (Invitrogen, Carlsbad, CA, USA)	Flow cytometry analysis and immunofluorescence	qRT-PCR	U6	[71]
miR-1233 ↑				81.0	76.0	n.a.					
miR-210 ↑	Diagnostic of ccRCC	Serum	45 ccRCC patients/30 healthy controls	82.5	80.0	n.a.	Total Exosome Isolation Reagent (from serum; Invitrogen, Carlsbad, CA, USA)	TEM ^e^; WB ^f^	qRT-PCR	miR-16-5p	[53]
Combination of miR-126-3p–miR-449a	Diagnostic of ccRCC	Urine	81 ccRCC patients/24 patients with benign lesions/33 healthy controls	60.6	100.0	0.820	Urine Exosome RNA Isolation Kit (Cat. 47200)	TEM	qRT-PCR	miR-16-5p + miR-106a-5p	[66]
Combination of miR-126-3p–miR-34b-5p				67.3	82.8	0.800					
Combination of miR-126-3p–miR-486-5p				52.9	95.8	0.790					
Combination of miR-25-3p–miR-34b-5p				73.1	79.3	0.760					
Combination of miR-21-5p–miR-34b-5p				74.0	72.4	0.760					
Combination of miR-150-5p–miR-126-3p				61.5	82.8	0.760					
miR-30c-5p ↓	Diagnostic biomarker of early-stage ccRCC	Urine	70 early-stage (T1aN0M0) ccRCC patients/30 early-stage prostate cancer (T1N0M0) patients/30 early-stage bladder cancer (T1N0M0) patients/30 healthy controls	68.57	100.0	0.819	Ultracentrifugation	NTA ^g^; TEM	qRT-PCR	Not Specified	[70]

^a^ AUC—area under the curve; ^b^ REF—reference; ^c^ ccRCC—clear cell renal cell carcinoma; ^d^ n.a.—not applicable; ^e^ TEM—transmission electron microscopy; ^f^ WB—Western blot; ^g^ NTA—nanoparticle tracking analysis. ↑ means upregulated and ↓ means downregulated.

**Table 4 cancers-13-05252-t004:** Promising prognostic exosomal miRNA in renal cell carcinoma (RCC).

Biomarker	Clinical Application	Source	Number of Cases/Controls	Consequences	Isolation	Characterization	Quantification Technique	Normalizer	REF ^a^
miR-224 (high levels)	Prognostic	Serum	108 ccRCC ^b^ patients	↓ OS ^c^, CSS ^d^, PFS ^e^	Total Exosome Isolation kit (from serum) (Invitrogen, Waltham, MA, USA)	TEM ^f^; WB ^g^	qRT-PCR	miR-16-5p	[74]
miR-26a-1-3p, miR- let-7i-5p, miRN-615-3p (high levels)	Prognostic	Plasma	Screening Phase: 40 ccRCC, 2 pRCC ^h^, 2 unspecified Validation Phase: 52 ccRCC, 2 chRCC ^i^, 6 pRCC, 5 unspecified	↑ OS	ExoQuick (System Biosciences, Mountain View, CA, USA	n.a. ^j^	qRT-PCR	miR-127-3p	[72]

^a^ REF—reference; ^b^ ccRCC—clear cell renal cell carcinoma; ^c^ OS—overall survival; ^d^ CSS—cancer-specific survival; ^e^ PFS—progression-free survival; ^f^ TEM—transmission electron microscopy; ^g^ WB—Western blot; ^h^ pRCC—papilar renal cell carcinoma; ^i^ chRCC—chromophobe renal cell carcinoma; ^j^ n.a.—not applicable. ↑ means longer survival and ↓ means shorter survival.

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
