# Peer review of "Unveiling the World of Circulating and Exosomal microRNAs in Renal Cell Carcinoma"

_cancers, 2021, doi:10.3390/cancers13215252_

Round 1

Reviewer 1 Report

Authors answered all my questions. However, I still think that the authors are not very clear on the mechanism behind microRNAs.

I recommend studying it thoroughly before writing a review.

However, also given the positive comments of the other reviewer, I think that the article can be accepted for publication.

This manuscript is a resubmission of an earlier submission. The following is a list of the peer review reports and author responses from that submission.

Round 1

Reviewer 1 Report

Sequeira et al. write a review of micro-RNA in renal cell carcinoma. Unfortunately the work is not concluded

Just for example: 

  • In many many micro-RNA authors didn't report the isoform (-5p, -3p)
  • They never report if micro-RNA are downregulated or upregulate
  • I think that authors must review the function of the micro-RNA reported.

In conclusion I think that the work is not publishable in the present state.

Reviewer 2 Report

The paper put forwards the advantage of Liquid Biopsies for cancer detection. 

I found the paper to be overall concise and easy to read.